# A genome-wide functional genomics approach uncovers genetic determinants of immune phenotypes in type 1 diabetes

Xiaojing Chu[1,2,3†], Anna WM Janssen[4†], Hans Koenen[5†], Linzhung Chang[1], Xuehui He[5], Irma Joosten[5], Rinke Stienstra[4,6], Yunus Kuijpers[2,3], Cisca Wijmenga[1], Cheng-Jian Xu[2,3,4], Mihai G Netea[4,7], Cees J Tack[4*], Yang Li[1,2,3,4*]

[1]Department of Genetics, University of Groningen, University Medical Center Groningen, Groningen, Netherlands; [2]Centre for Individualised Infection Medicine, CiiM, a joint venture between the Hannover Medical School and the Helmholtz Centre for Infection Research, Hannover, Germany; [3]TWINCORE, Centre for Experimental and Clinical Infection Research, a joint venture between the Hannover Medical School and the Helmholtz Centre for Infection Research, Hannover, Germany; [4]Department of Internal Medicine, Radboud University Medical Center, Nijmegen, Netherlands; [5]Department of Laboratory Medicine, Laboratory Medical Immunology, Radboud University Medical Center, Nijmegen, Netherlands; [6]Division of Human Nutrition and Health, Wageningen University, Wageningen, Netherlands; [7]Department for Genomics & Immunoregulation, Life and Medical Sciences Institute (LIMES), University of Bonn, Bonn, Germany

*For correspondence:
Cees.Tack@radboudumc.nl (CJT);
yang.li@helmholtz-hzi.de (YL)

†These authors contributed equally to this work

Competing interest: The authors declare that no competing interests exist.

## Abstract

**Background:** The large inter-individual variability in immune-cell composition and function determines immune responses in general and susceptibility o immune-mediated diseases in particular. While much has been learned about the genetic variants relevant for type 1 diabetes (T1D), the pathophysiological mechanisms through which these variations exert their effects remain unknown.

**Methods:** Blood samples were collected from 243 patients with T1D of Dutch descent. We applied genetic association analysis on >200 immune-cell traits and >100 cytokine production profiles in response to stimuli measured to identify genetic determinants of immune function, and compared the results obtained in T1D to healthy controls.

**Results:** Genetic variants that determine susceptibility to T1D significantly affect T cell composition. Specifically, the CCR5+ regulatory T cells associate with T1D through the CCR region, suggesting a shared genetic regulation. Genome-wide quantitative trait loci (QTLs) mapping analysis of immune traits revealed 15 genetic loci that influence immune responses in T1D, including 12 that have never been reported in healthy population studies, implying a disease-specific genetic regulation.

**Conclusions:** This study provides new insights into the genetic factors that affect immunological responses in T1D.

**Funding:** This work was supported by an ERC starting grant (no. 948207) and a Radboud University Medical Centre Hypatia grant (2018) to YL and an ERC advanced grant (no. 833247) and a Spinoza grant of the Netherlands Association for Scientific Research to MGN CT received funding from the Perspectief Biomarker Development Center Research Programme, which is (partly) financed by the Netherlands Organisation for Scientific Research (NWO). AJ was funded by a grant from the European Foundation for the Study of Diabetes (EFSD/AZ Macrovascular Programme 2015). XC was supported by the China Scholarship Council (201706040081).

## Editor's evaluation

This study examines genetic and non-genetic factors influencing immune responses in type 1 diabetes Key findings are: 1) age and season affect immune cell traits and cytokine production upon stimulation; 2) certain genetic variants that determine susceptibility to T1D significantly affect T cell composition, notably the CCR region that is associated with CCR5+ regulatory T cells; and 3) 15 genetic loci that influence immune responses in T1D, most of which have not been seen previously in healthy populations. The results suggest mechanisms of T1D-specific genetic regulation.

## Introduction

Type 1 diabetes (T1D) is a common, chronic, autoimmune disease, characterized by destruction of insulin-producing beta-cells in the pancreas that results in lifelong dependence on exogenous insulin and is associated with a high morbidity and mortality (*Atkinson et al., 2014*). The causes and immunological pathways responsible for T1D development are still incompletely understood, which hampers the efforts to identify an etiopathogenetic treatment.

Many studies have highlighted the role of environmental, genetical, and immunological factors in the pathogenesis of T1D (*Pociot and Lernmark, 2016*; *Rewers and Ludvigsson, 2016*). Environmental factors such as being overweight, infections, microbiome composition, and dietary deficiencies have been reported as risk factors for T1D (*Rewers and Ludvigsson, 2016*). In turn, the immunological pathogenesis (*Cabrera et al., 2016*) of T1D includes innate inflammation and adaptive immunity, such as enhanced T cell responses (*Hundhausen et al., 2016*). In the last two decades, large genome-wide association studies (GWAS) performed have underscored the contribution of genetic polymorphisms to T1D for the susceptibility, with ~60 genomic loci associated with T1D risk identified (*Barrett et al., 2009*; *Bradfield et al., 2011*; *Cooper et al., 2008*; *Grant et al., 2009*; *Huang et al., 2012*; *Onengut-Gumuscu et al., 2015*; *Ram et al., 2016*). While these loci show significant enrichment in specific immune-related biological pathways, such as cytokine signaling and T cell activation (*Barrett et al., 2009*; *Cooper et al., 2008*), the functional consequences of many of these loci and genetic variants are still unknown. We thus lack information that could link the genetic susceptibility factors to the immunological pathways potentially important for T1D pathogenesis. The genetically regulated inflammatory response signature in T1D may also be relevant for the inflammatory response in general and may become modified by the chronic hyperglycemic state.

In the present study we aimed to comprehensively describe the immunopathological consequences of the genetic variants linked to T1D susceptibility, using a high-throughput functional genomics approach. As a part of the Human Functional Genomics Project (HFGP) (*Netea et al., 2016*), we carried out deep immunophenotyping in peripheral blood samples from a cohort of 243 T1D patients (300DM) using cell subpopulation composition and cytokine production upon stimulations as proxies of immunological function. Part of the results were then compared to those obtained in a population-based cohort of 500 healthy individuals (500FG) that successfully characterized the impact of genetic factors (*Aguirre-Gamboa et al., 2016*; *Li et al., 2016*) on immune responses in healthy individuals. Here, we systematically evaluate the genetic regulation of the immune phenotypes in T1D and show how genetic variations affect immune-cell traits and cytokine production in response to stimulations. In total, we identify 15 genome-wide significant genomic loci (p-value $< 5 \times 10^{-8}$) associated with immune phenotypes in the 300DM cohort, including 12 novel loci that have never been reported in any healthy population study. These data provide a deeper understanding of the immune mechanisms involved in the pathophysiology of T1D and affecting the general inflammatory response and may open avenues toward the development of novel diagnostics and potentially immunotherapies.

## Materials and methods
### Study cohort

This study mainly focuses on a 300DM cohort, and involves a 500FG cohort (part of HFGP; *Netea et al., 2016*). In the 300DM cohort, we collected blood samples from 243 T1D patients (132 males and 111 females of Caucasian origin), following a previously described methodology (*Aguirre-Gamboa et al., 2016*; *Ter Horst et al., 2016*; *Li et al., 2016*). Participant's ages ranged from 20 to 84 years.

**eLife digest** Every year around the world, over 100,000 people are diagnosed with type 1 diabetes. This disease develops when the immune system mistakenly destroys the cells that produce a hormone called insulin, leaving affected individuals unable to regulate their blood sugar levels. Type 1 diabetes patients must rely on regular injections of manufactured insulin to survive.

The composition and activity of the human immune system is under genetic control, and people with certain changes in their genes are more susceptible than others to develop type 1 diabetes. Previous studies have identified around 60 locations in the human DNA (known as loci) associated with the condition, but it remains unclear how these loci influence the immune system and whether diabetes will emerge.

Chu, Janssen, Koenen et al. explored how variations in genetic information can influence the composition of the immune system, and the type of molecules it releases to perform its role. To do so, blood samples from 243 individuals of Dutch descent with type 1 diabetes were collected, and genetic associations were investigated.

The results revealed that a major type of immune actors known as T cells are under the control of genetic factors associated with type 1 diabetes susceptibility. For instance, a specific type of T cells showed shared genetic control with type 1 diabetes. In addition, 15 loci were identified that influenced immune responses in the patients. Among those, 12 have never been reported to be involved in immune responses in healthy people, implying that these regions might only regulate the immune system of individuals with type 1 diabetes and other similar disorders. Finally, Chu, Janssen, Koenen et al. propose 11 genes within the identified loci as potential targets for new diabetes medication. These results represent an important resource for researchers exploring the genetic and immune basis of type 1 diabetes, and they could open new avenues for drug development.

Detailed information about the 500FG cohort can be found in the previous publications (*Aguirre-Gamboa et al., 2016*; *Ter Horst et al., 2016*; *Li et al., 2016*).

## Measurement of immune-cell composition

Myeloid and lymphoid immune-cell levels were measured by 10-color flow cytometry, and we calculated the parental and grandparental proportion of 73 manually annotated immune cells and the proportion of CD4+ T cells, CD8+ T cells, memory Tregs and monocytes carrying the chemokine receptors: CCR6 (CD196), CXCR3 (CD183), CCR4 (CD194), CCR5 (CD195), and CCR7 (CD197). In total, we ended up with 269 immune-cell traits for the 300DM cohort. We then used the same gating strategy of measuring cell subpopulations as 500FG (see *Aguirre-Gamboa et al., 2016* and *Figure 1—figure supplement 1*).

## Stimulation of PBMCs and measurement of cytokine production capacity

Isolated peripheral blood mononuclear cells (PBMCs) were washed twice with cold phosphate-buffered saline and suspended in Roswell Park Memorial Institute (RPMI) 1640 Dutch-modified culture medium (Gibco/Invitrogen, Breda, the Netherlands) supplemented with 50 mg/l gentamycin (Centraform), 1 mM pyruvate (Gibco/Invitrogen), and 2 mM L-glutamine (Gibco/Invitrogen). Cells were counted on a Sysmex XN-450 Hematology Analyzer (Sysmex Corporation, Kobe, Japan).

For the in vitro stimulation experiments, $5 \times 10^5$ cells/well were cultured for 24 hr or 7 days at 37°C and 5% $CO_2$ in 96-well round-bottom plates (Greiner). For the 7-day cultures, the medium was supplemented with human pooled serum (pooled from healthy blood donors, end concentration 10%). Supernatants were collected and stored in –20°C until used for ELISA.

The following stimulations were used: LPS (100 ng/ml, 1 ng/ml), Pam3cys, *Borrelia* mix, *Candida albicans* conidia, *C. albicans* hyphae, imiquimod (IMQ), *Staphylococcus aureus*, *Streptococcus pneumoniae*, palmitic acid (C16), C16+ monosodium urate crystals, *Escherichia coli*, *Mycobacterium tuberculosis*, *Borrelia burgdorferi*, *Coxiella burnetii*, *Cryptococcus neoformans*, oxidized low-density lipoprotein (OxLDL), OxLDL+ LPS, polyinosinic:polycytidylic acid, and *Rhizopus* microspores, *Rhizopus oryzae* (*Li et al., 2016*). Concentrations of cytokines in response to various stimulations

were measured in PBMCs by ELISA kits, following the manufacturer's protocol. Following our previously reported definition (*Li et al., 2016*), we considered IL-1β, IL-6, TNF-α, IL-8, IL-10, IL-1Ra, and MCP-1 to be monocyte-derived cytokines and measured them 24 hr after stimulation. Likewise, we considered INF-γ, IL-17, and IL-22 to be T cell-derived cytokines and measured them at 7 days after stimulation. For all cytokines commercially available kits were used (R&D Systems, Minneapolis, MN, or Sanquin, Amsterdam, the Netherlands).

## Genotyping, quality control, and imputation

DNA samples of 224 Dutch T1D patients were collected and genotyped using the Infinium Global Screening Array. Genotype calling was performed using Opticall 0.7.0 (*Shah et al., 2012*) with default settings (call rate > 0.99). We then excluded two samples due to contamination or mislabeling, one sample from a related individual (identity be descent > 0.185) and six genetic outliers identified by either heterozygosity rate check (individuals with heterozygosity rate heterozygosity rate ±3 SD from the mean were excluded) or multidimensional scaling (MDS) plots of samples merging with 1000 Genomes data. This left 215 samples for further analysis, and these samples show good consistency with European samples from 1000 Genomes data (*Figure 1—figure supplement 2*) according to standard protocol (*Anderson et al., 2010*). DNA samples from the 500FG cohort were genotyped by Illumina Human OmniExpress Exome-8 v1.0 single-nucleotide polymorphism (SNP) chip. Outliers were excluded according to population relationship, medication, and disease information. Details can be found in our previous studies in 500FG (*Aguirre-Gamboa et al., 2016*; *Li et al., 2016*). SNPs with a minor allele frequency (MAF) <0.001 were removed from each cohort, and samples with a Hardy-Weinberg equilibrium (HWE) p-value $< 1 \times 10^{-5}$ were removed for the healthy cohort (500FG). We merged the 300DM cohort and 500FG cohort by taking shared genetic variants. We then imputed their genotypes using an online genotype imputation service provided by Michigan Imputation Server (https://imputationserver.sph.umich.edu)(*Das et al., 2016*) with HRC Panel 1.1 as reference. We excluded SNPs with low imputation quality (R2 < 0.3) and/or a MAF < 0.01 in all imputed samples and/or HWE p-values $< 1 \times 10^{-5}$ in healthy individuals, leaving 4,304,387 SNPs and 666 individuals in the merged cohort ($N_{300DM}$ = 215 and $N_{500FG}$ = 451).

## Preprocessing the data

All statistical analyses were performed using the statistical programming language R. Immune-cell proportion was calculated by dividing counts of their parental by grandparental cell types (*Supplementary file 1A*). An inverse rank transformation (*Aguirre-Gamboa et al., 2016*; *Orrù et al., 2013*) was applied on the proportion values for genetic association analysis. Cytokine levels were log2-transformed.

## Immune parameter quantitative trait locus mapping

After intersecting with available genotype data and excluding volunteers with a mixed background or other genetic background, 214 samples were left for immune parameter quantitative trait locus (QTL) mapping. We next evaluated covariates influencing immune function. Associations with age, gender, and seasonal effects were calculated following previously described methods (*Aguirre-Gamboa et al., 2016*; *Ter Horst et al., 2016*; *Li et al., 2016*), in short, using Spearman's correlation analysis and a linear regression model. Significance was declared after multiple-testing correction (FDR < 0.05). Age significantly associated with 129/269 immune-cell traits and 12/55 cytokine traits, gender associated with 59/269 immune-cell traits and 5/55 cytokine traits, and seasonal effects associated with 121/269 immune-cell traits and 38/55 cytokine traits (*Figure 4—figure supplement 1*). We therefore took age, gender, and seasonal effects as covariates in the linear regression for both immune-cell proportion QTL mapping and cytokine QTL mapping. In addition, considering the effect of immune-cell proportion on cytokine production, we took major cell types, including monocyte, lymphocyte, T cell, B cell, and NK cell proportion in PBMC as covariates in a linear model for cytokine QTL mapping, as we have done previously (*Li et al., 2016*). Linear regression was applied using the R package Matrix-eQTL (*Shabalin, 2012*) for immune parameter QTL mapping, and the software METAL (*Willer et al., 2010*) was applied to summary statistics for both cohorts for a meta-analysis, in which the model based on effect size and standard error with default settings were used. p-Values $<5 \times 10^{-8}$ were considered to be genome-wide significant. Calculated lambda values indicated no obvious inflations (0.977–1.026).

### Extraction of the T1D GWAS SNP list

We downloaded a summary of T1D GWAS results from the GWAS-catalog (https://www.ebi.ac.uk/gwas/) (*Buniello et al., 2019*) in November 2019, and removed data from studies performed in non-European-ancestry populations. Top SNPs from different studies that were in LD ($r^2 > 0.1$) were considered as the same locus, and the SNPs with the lowest p-value were included in the analysis. We noticed that the effect directions of some SNPs were unclear or inconsistent between different studies. In this case, we assigned the direction from the most-recent GWAS (*Onengut-Gumuscu et al., 2015*).

### GWAS analysis of the 300DM cohort for the known T1D loci

We extracted all proxies in strong LD with the top SNPs from published T1D GWAS (case-control) studies ($r^2 > 0.8$) and performed a chi-square test on clinical status by using PLINK 1.9. Samples in 300DM were taken as cases and samples in 500FG as controls.

### Impact of T1D GWAS loci on immune phenotypes

To detect the impact of T1D GWAS loci on immune-cell populations, we grouped all traits into four categories (B cells, T cells, monocytes, and NK cells), and counted the number of suggestive associations (p-value < 0.05) between the 63 top SNPs from T1D GWAS loci and immune-cell traits. 1000 permuted sets of the 63 SNPs were randomly selected from independent SNPs ($r^2 > 0.2$) pruned from all genotyped SNPs. We then compared the associations of the 63 top GWAS SNPs with the associations between the 1000 permutated sets and the same category of immune traits.

We further applied a multivariate linear model to estimate the proportion of variance of each immune phenotype explained by the top SNPs from T1D GWAS loci. We repeated this analysis on 1000 permuted sets of the 63 independent SNPs, which were used as reference set. We then compared the null distribution with the variance explained by 63 top SNPs from T1D GWAS loci. The p-value was calculated by the percent of explained variance from permuted sets greater than the variance explained by the 63 T1D GWAS SNPs.

### Gene expression analysis on *CCR5* and its corresponding ligands genes

Normalized gene expression data in PBMCs and pancreas from T1D and controls were acquired from https://www.ebi.ac.uk/gxa (*Papatheodorou et al., 2020*). A Student's t-test was applied to compare gene expression between groups.

### Post-QTL analysis

An R package ggplot2 was used to generate Manhattan plots and boxplots. Locus zoom plots were made using an online tool (http://locuszoom.org; *Pruim et al., 2010*). We used R package coloc (*Giambartolomei et al., 2014*) to perform colocalization analysis with T1D GWAS summary statistics and immune parameter QTL profiles. For pathway analysis, genes located within ±10 kb of genome-wide significant SNPs (p-value < $5 \times 10^{-8}$) were extracted and analyzed using the FUMA online tool (*Watanabe et al., 2017*) (https://fuma.ctglab.nl). ANNOVAR was used to annotate genetic variants (*Wang et al., 2010*).

### Other packages used in this paper

Pheatmap was used to make heatmaps. Scatter plots and bar plots were generated using ggplot2.

## Results

### Interrelationship between immune-cell counts and cytokine production in T1D

We collected blood samples from 243 T1D patients (300DM cohort), following a previously described methodology (*Aguirre-Gamboa et al., 2016*; *Ter Horst et al., 2016*; *Li et al., 2016*). The baseline characteristics of the 300DM and a cohort of healthy individuals (500FG) are shown in *Supplementary file 1B*. Their median age was 53.5 years (range 20–85), and they had a median diabetes duration of 28 years (range 1–71 years). Hence, the cohort generally consisted of middle-aged people with long-standing T1D. We measured 72 types of immune cells covering both lymphocytes and monocyte

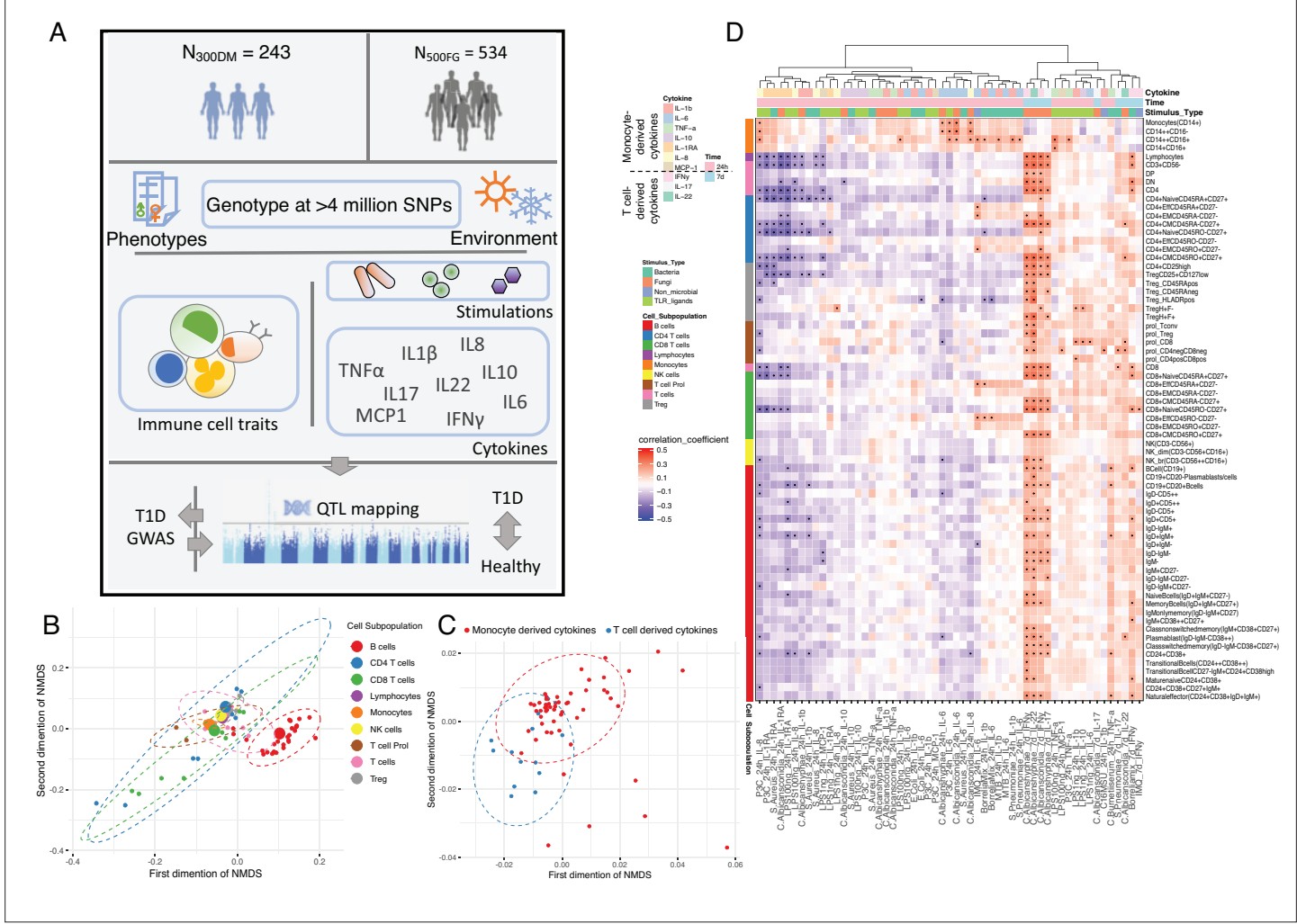

**Figure 1.** Overview of data and experimental design. (**A**) Schematic study design. (**B**,**C**) Cell-type (**B**) and cytokine-type (**C**) relationship visualized by nonmetric multidimensional scale (NMDS) analysis in 300DM. (**B**) Cell traits (red) cluster separately from T cell and monocyte traits (**B**) and cytokines with the same cellular origins cluster together (**C**). In panel (**B**), small dots indicate the proportion of subpopulations and large dots indicate counts of their parental cell types. Circles represent the calculated centroid of the grouped cell and cytokine types at confidence level 0.95. (**D**) Heatmap of correlation coefficients between immune-cell counts (y-axis) and cytokine production in response to stimulations (x-axis) in T1D patients. Significant correlations (FDR < 0.05) are labeled by dots, with color indicating correlation coefficients. Positive correlations (red) are observed between monocyte traits and monocyte-derived cytokines (top left), and between adaptive immune-cell counts and T cell-derived cytokines (bottom right). Negative correlations (blue) are observed between adaptive immune-cell counts and monocyte-derived cytokines.

The online version of this article includes the following figure supplement(s) for figure 1:

**Figure supplement 1.** Heterozygosity check (left, x-axis: Proportion of missing genotypes, y-axis: Heterozygosity rate) and ancestry clustering (right, C1: component 1, C2: component 2 in multiple dimensional scaling plot).

**Figure supplement 2.** Flow cytometry gating strategy of the chemokine receptor panel.

**Figure supplement 3.** Correlation between cell counts and cytokine production in healthy individuals.

lineages and 10/6 (300DM/500FG) different cytokines released in response to stimulation with four types of human pathogens in both cohorts (***Figure 1A***).

Nonmetric MDS plots illustrate the interrelationship among immune-cell abundances (***Figure 1B***) and cytokine production levels (***Figure 1C***) in T1D patients. We observed a separation of the B cell subpopulations cluster from the T cell, monocyte, and NK cell population clusters. This suggests that T cells, monocytes, and NK cells have more interplay than B cells at baseline, which is consistent with

our previous finding in a healthy cohort (500FG; *Aguirre-Gamboa et al., 2016*). Furthermore, cytokine features are clustered based on their cellular origins, with a partially overlapping cluster between monocyte-derived cytokines and T cell-derived cytokines (*Figure 1C*). This may suggest activation of a co-regulatory network of monocyte-derived and T cell-derived cytokine production capacity in T1D.

To obtain a comprehensive interaction map between immune cells and cytokines, we correlated each of the immune-cell counts with each of the cytokine production profiles (IL-1β, IL-6, TNF-α, IL-10, IL-1RA, IL-8, MCP-1, IFNγ, IL-17, and IL-22) in response to 21 stimulations. Cytokine levels were hierarchically clustered based on correlation coefficients with immune-cell counts (*Figure 1D*). In line with the functional relationship between immune cells and cytokines, we observed positive correlations between monocyte lineages and monocyte-derived cytokines (IL-1β, IL-6, TNF-α, IL-10, IL-1RA, IL-8, and MCP-1) and between T cell subsets and T cell-derived cytokines (IFNγ, IL-17, and IL-22) (indicated by red color in *Figure 1D*). We also found a negative correlation between monocyte-derived cytokines production in response to four distinct types of stimulations (bacteria, fungi, non-microbial, and TLR ligands) and lymphocyte counts (blue, *Figure 1D*). We observed similar correlation patterns in 500FG (*Figure 1—figure supplement 3*). This correlation is in line with a previous finding that a high abundance of adaptive immune cells at baseline is associated with lower production of monocytes-derived cytokines after stimulation (*Kim et al., 2007*), and this is not altered by T1D status. Overall, the interrelationships between immune cells and cytokines in response to stimulations are roughly similar between T1D and healthy individuals.

## Impact of T1D GWAS SNPs on immune phenotypes in T1D patients

Considering that T1D is a multifactorial disease with a genetic component, we tested whether the known risk variants of T1D affect immune phenotypes and function. We first checked SNPs within the HLA locus in our association studies on cell proportion and cytokine production level. Consistent with our previous findings in 500FG, we did not observe any significant associations of HLA allelic variants in 300DM. We then acquired non-HLA genetic loci from published GWAS of European background were acquired from the GWAS-catalog (November 2019)(*Buniello et al., 2019*). Among these, genetic variants in 63 independent T1D loci were present in our data, and we found that 13 of these 63 were indeed associated with susceptibility to T1D with nominal significance (p-value < 0.05) (*Supplementary file 1C*).

We next investigated whether these genetic risk loci for T1D affect immune parameters and function. The quantile-quantile plot of the association of the 63 T1D GWAS loci with different cell types and cytokines illustrates an inflated deviation from an expected uniform distribution (*Figure 2A*, *Figure 2—figure supplement 1*). We further tested whether this deviation can be explained by chance by comparing the association of immune traits with T1D GWAS SNPs with that of 1000 randomly selected independent SNPs (*Figure 2B*, Materials and methods). The p-value shows that the T1D GWAS SNPs are enriched in association with T cell traits in the T1D cohort (p-value = 0.007).

A pair-wise association analysis between T1D GWAS loci and immune phenotypes shows that 261 out of 269 immune-cell phenotypes and 53 out of 55 cytokine-stimulation pairs are suggestively associated with at least one T1D GWAS locus (p-value < 0.05, *Supplementary file 1d and e*). We further applied a permutation-based approach to test whether immune phenotypes were significantly influenced by the cumulative effects of these 63 GWAS loci (Materials and methods). Compared to random sets of independent SNPs, the 63 T1D GWAS loci explain significantly more of the variance in 27 cell sub-proportions and 15 cytokine production traits (p-value < 0.05, *Figure 2C*, *Figure 2—figure supplement 2A, B*,). As shown in the heatmap (*Figure 2C*, arrowhead), one T1D risk allele, rs11574435-T, which is in strong LD ($r^2$=0.95) with the T1D GWAS SNP rs113010081 (*Onengut-Gumuscu et al., 2015*), is associated with a higher percentage of many CCR5+ CD4+ T cell traits and a lower percentage of CCR5– CD4+ T cell traits.

Chemokine signaling pathways regulate the migration of cells from the circulation (PBMCs) to the tissue (pancreas). To further validate the importance of chemokine signaling mediated by *CCR5* in T1D, we illustrated the transcriptional changes on *CCR5* and its corresponding ligand genes using publicly available data from transcriptome analysis in PBMCs and pancreatic tissue from T1D patients and controls (*Planas et al., 2010*; *Yang et al., 2015*) (see also Materials and methods). This identified significant expression changes of *CCR5*, *CCL5,* and *CCL4* in T1D patients, which suggests the involvement of this chemokine ligand-chemokine receptor pathway (*Figure 3A*). In addition, another

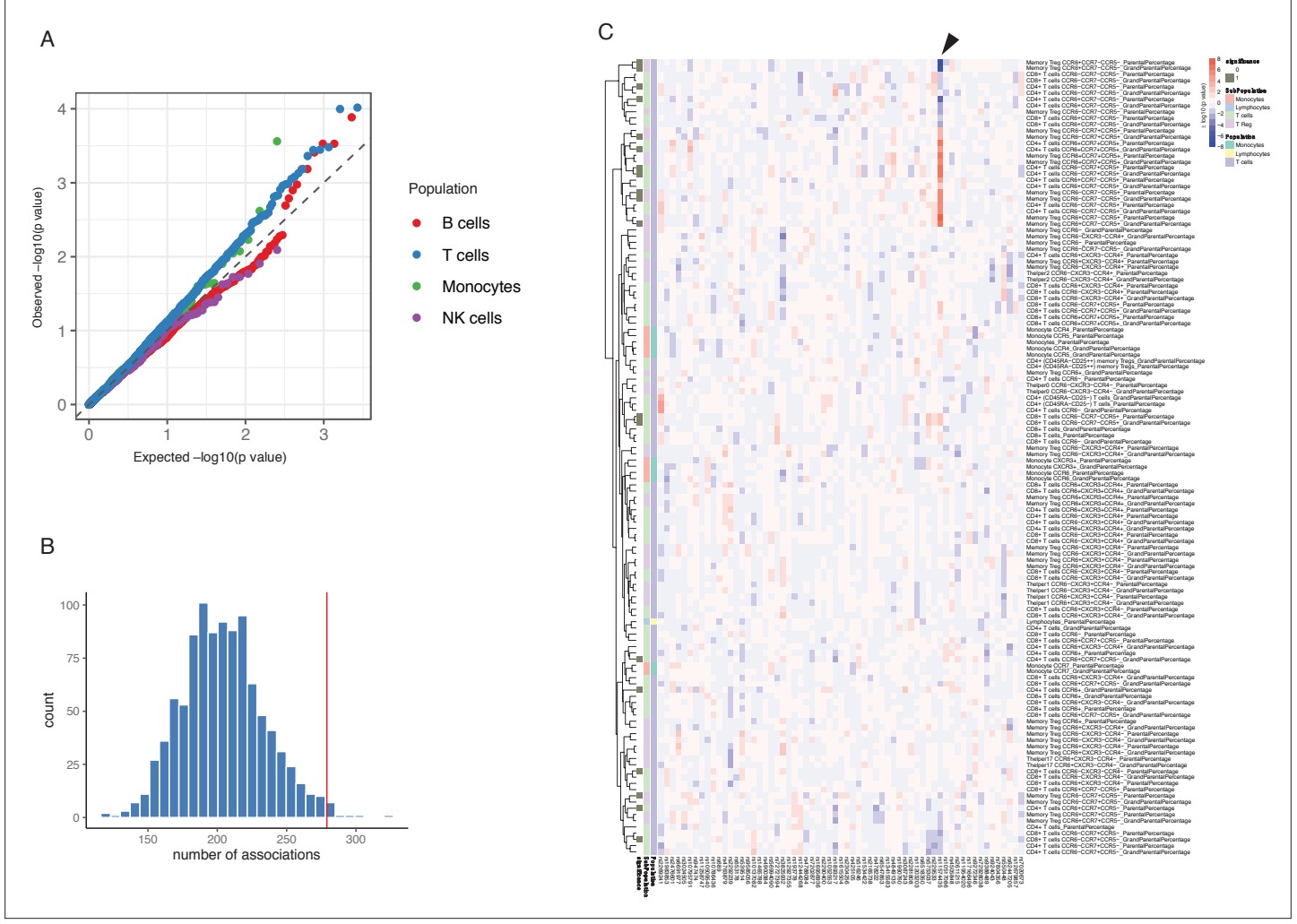

**Figure 2.** Impact of type 1 diabetes (T1D) genome-wide association studies (GWAS) single-nucleotide polymorphisms (SNPs) on immune phenotypes. (**A**) Quantile-quantile (Q-Q) plots of quantitative trait locus (QTL) profiles of 62 T1D GWAS loci grouped by cell populations. The distribution of p-values of associations with T cells traits (blue) shows a significant deviation from an expected uniform distribution (dashed line). (**B**) Histogram showing number of associations observed (red line) and those in permutations (blue bars). (**C**) Heatmap of QTL profiles of cell proportion carrying certain chemokine receptors across 62 T1D GWAS loci, colored by −log10(p-values) and effect direction of the T1D risk allele. Arrowhead indicates a T1D risk allele rs11574435-T.

The online version of this article includes the following figure supplement(s) for figure 2:

**Figure supplement 1.** Qqplots of QTL profiles of 62 T1D GWAS loci grouped by cytokine types.

**Figure supplement 2.** Impact of T1D GWAS SNPs on immune phenotypes.

top SNP, rs35092096, within the *CCR* gene region has the strongest effects on many CCR5 Treg proportions in T1D. For example, minor allele T in rs35092096 associates with a higher ratio of CCR6+ CCR7− CCR5+ Tregs/CCR6+ Tregs (*Figure 3B and C*). Together with the observed regulation from a GWAS locus within the *CCR* region on CCR6+ CCR7− CCR5+ Treg proportion, we tested whether CCR6+ CCR7− CCR5+ Tregs and T1D share the same causal variants/genomic regions by integrating the cell proportion QTL of CCR6+ CCR7− CCR5+ Tregs and the latest T1D GWAS profile via colocalization analysis (*Giambartolomei et al., 2014*). The result strongly supports that CCR6+ CCR7− CCR5+ Tregs share the same regulatory genomic region with T1D, although the causal SNP might be different (*Figure 3D*, H3=0.95). Altogether, these results support a role for the *CCR* region in Treg function in the pathogenesis of T1D.

Overall, we observe that T1D GWAS loci influence immune-cell proportion and cytokine production capacity, again stressing the importance of T cell immunity in genetic regulation of T1D.

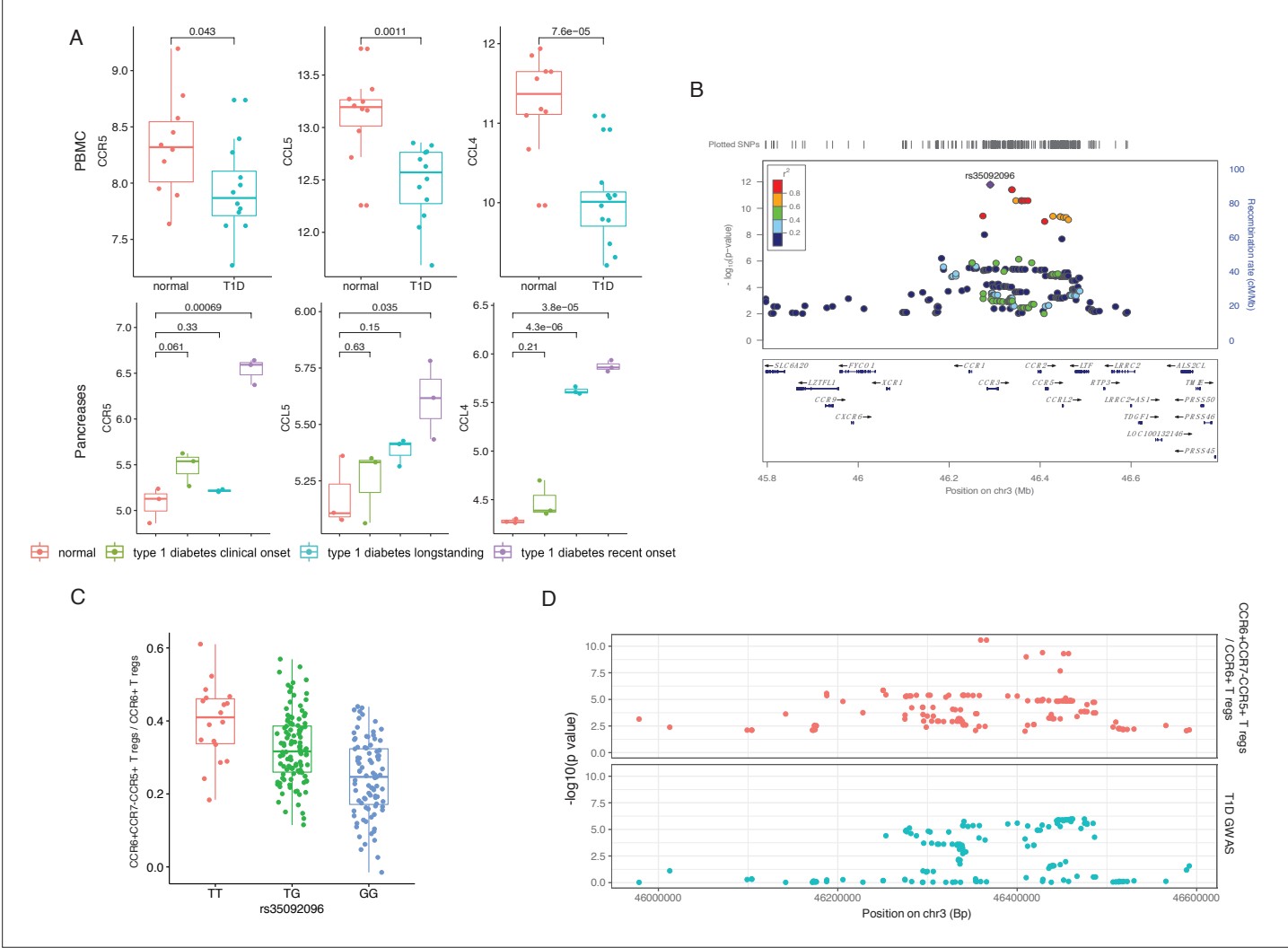

**Figure 3.** Genetic impact on CCR5-mediated chemokine signaling in type 1 diabetes (T1D). (**A**) Expression of *CCR5, CCL5,* and *CCL4* in peripheral blood mononuclear cells (PBMCs) (top) and pancreas (bottom) is altered in T1D patients (blue, green, and purple) compared to healthy controls (red). (**B**) Locus zoom plot showing that single-nucleotide polymorphisms (SNPs) around rs35092096 located in the CCR region are associated with CCR6+ CCR7− CCR5+ Treg proportion. (**C**) Boxplot showing that CCR6+ CCR7− CCR5+ Treg proportion differs in different rs35092096 genotypes (TT: red, TG: green, and GG: blue). (**D**) Two locus zoom plots showing colocalization between CCR6+ CCR7− CCR5+ Treg proportion quantitative trait locus (QTL) profiles (top, red) and T1D genome-wide association studies (GWAS) profile (bottom, blue) within CCR regions.

## Genetic regulation of immune phenotypes in T1D

To further explore potential genetic regulation of immune phenotypes on the whole-genome level, we performed QTL mapping in 300DM. This identified nine genome-wide significant QTLs (p-value $< 5 \times 10^{-8}$) associated with immune-cell proportion, including four associated with T cell subpopulations expressing specific chemokine receptors (e.g., rs35092096 and rs7614884) (*Figure 4A*, top and middle panels, *Supplementary file 1F*). Pathway analysis of the cell proportion QTLs showed significant enrichment in chemokine and cytokine signaling-related biological pathways (FDR < 0.05, *Figure 4—figure supplement 2A*), highlighting the effects of immune signaling genes in cell proportion regulation.

In parallel, we detected six significant genomic loci associated with cytokine production in response to stimulations by QTL mapping in 300DM (*Figure 4A*, bottom panel, *Supplementary file 1F*). Cytokine production QTLs are significantly enriched in TLR-related pathways (FDR < 0.05, *Figure 4—figure supplement 2*). A missense variant, rs5743618, on the *TLR1* gene, affects IL-6 production in response to *S. pneumoniae* in PBMC. Previously, we had observed that the TLR locus has a strong

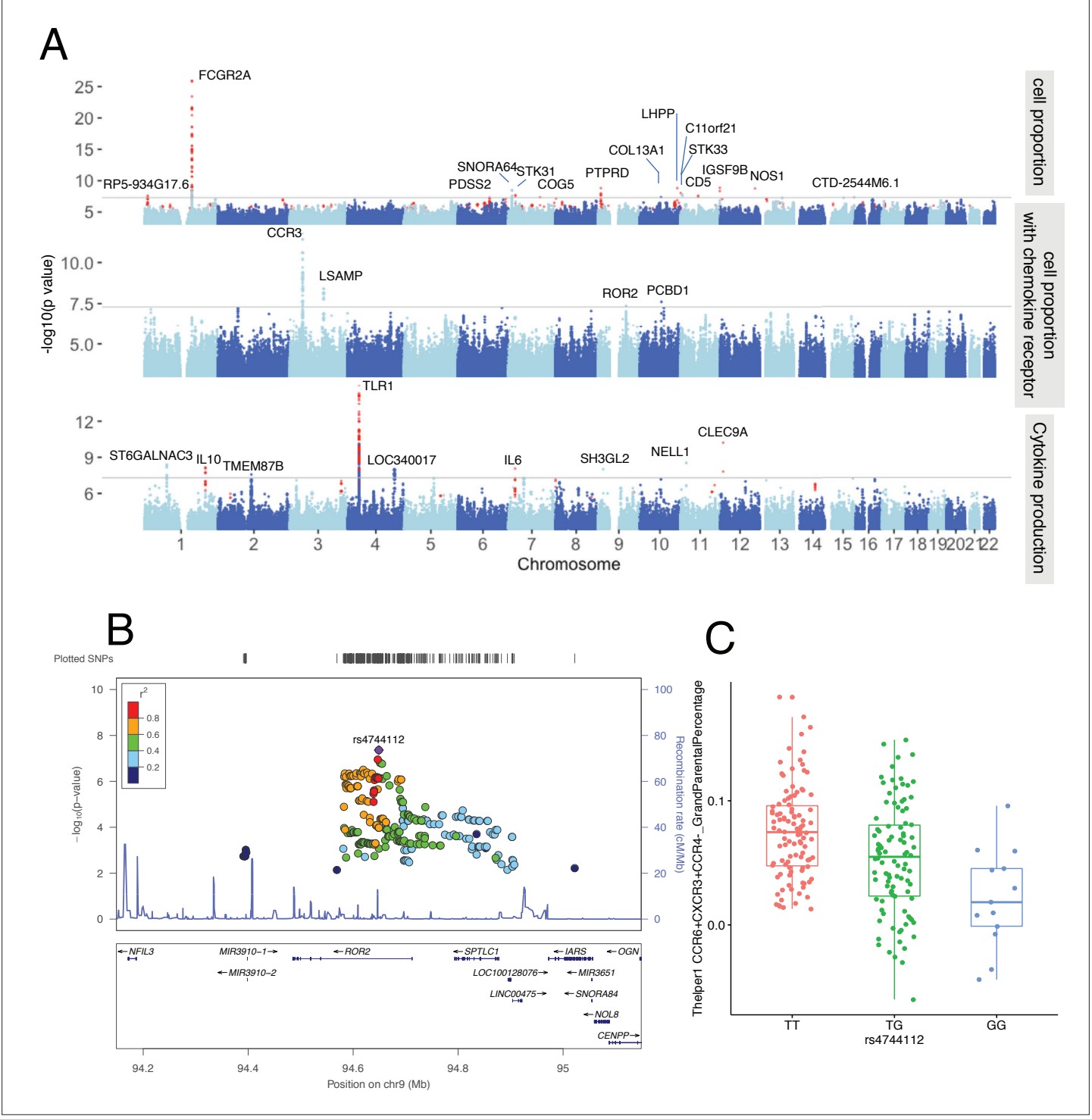

**Figure 4.** Genetic regulators of immune phenotypes. (**A**) Three Manhattan plots showing genetic regulators of immune-cell proportion (top), proportion of immune cells expressing CCR (middle) and cytokines production in response to stimulations (bottom). p-Values of single-nucleotide polymorphisms (SNPs) identified in meta-analysis are colored red. (**B**) Locus zoom plot showing a type 1 diabetes (T1D)-specific regulatory locus around rs4744112 that effects CCR6+ CXCR3+ CCR4 T helper1 proportion. (**C**) Boxplot showing how CCR6+ CXCR3+ CCR4 T helper1 proportion varies in different rs4744112 genotypes (TT: red, TG: green, and GG: blue).

The online version of this article includes the following figure supplement(s) for figure 4:

**Figure supplement 1.** Enriched pathways of QTLs found for immune phenotypes.

*Figure 4 continued on next page*

*Figure 4 continued*

**Figure supplement 2.** Functional annotation of the immune phenotypes associated SNPs (P < 5 × 10-8).

**Figure supplement 3.** Impact of age, gender and seasons on immune phenotypes.

effect on cytokine levels in controls (*Li et al., 2016*). Our results support the reported function of TLRs in detecting pathogens (including bacterial pathogens) and triggering the release of inflammatory cytokines (*Fitzgerald and Kagan, 2020*).

Next, to identify the consistent immune phenotype QTLs within T1D patients and healthy individuals and expand our understanding of genetic regulation of immune phenotypes, we performed a meta-analysis by integrating the QTLs obtained from the 300DM and 500FG cohorts. In total, we identified 10 novel genetic loci associated with immune-cell proportion (*Figure 4A*, top panel, colored in red) and three novel genetic loci that were associated with cytokine production (*Figure 4A*, bottom panel, colored in red). This included a *CD5* locus that was significantly associated with IgD+ CD5++ proportion in B cells (*Supplementary file 1F*), which may be important as recent studies have reported the relevance of IgD+ CD5++ B cells in T1D (*Saxena et al., 2017*) and other autoimmune diseases such as Graves' disease (*Van der Weerd et al., 2013*).

To summarize, we identified 28 genetic loci associated with either immune-cell traits or cytokine production phenotypes. Among them, the genes within 11 loci (±250 kb window) are known drug targets according to the public drug database (*Finan et al., 2017*), including *COL13A1*, *CCR* family genes, *ROR2*, *NOS1*, *IL-10*, *IL-6*, *STK33*, *NELL1*, *FCGR2A*, *CD5*, and *TLR1* (*Supplementary file 1F*). Some of these genes are already therapeutic targets in other autoimmune diseases, for example, the *STK33* inhibitor for rheumatoid arthritis treatment (*Rolf et al., 2015*).

It is also worth mentioning that, as far as we know, 12 out of the 28 significant loci we identified have never been reported in other healthy individual population cohorts according to PhenoScanner V2 (*Kamat et al., 2019*) (p-value < 1 × 10$^{-5}$, November 2019). Nor were they identified in the 500FG cohort (p-value < 0.05). One example effect of one of these T1D-specific loci is that of rs4744112 located on chromosome 9, which influences CCR6+ CXCR3+ CCR4− Th1-like helper proportion (*Figure 4B*), with minor allele G leading to a decrease in these cells relative to major allele T (*Figure 4C*). rs4744112 is located within the transcriptional starting site and enhancer regions (*Kundaje et al., 2015*) of *ROR2*.

### Functional clues from the associated variants in T1D patients

To understand the mechanism behind the genetic regulation of immune response in T1D, we next explored the function of the identified genetic factors behind immune phenotypes. We noticed that SNPs within the 28 immune parameter QTLs are mostly located in intergenic and intronic regions (*Figure 4—figure supplement 3*), suggesting that the genetic variants we identified influence immune phenotypes through regulatory effects rather than by altering protein structure alteration. Moreover, according to public databases (*Carithers and Moore, 2015*; *Westra et al., 2013*), 7 out of the 28 loci influence gene expression in blood (*Supplementary file 1F*), including rs7512140: FCGR2B/FCRLB, rs35092096: CCR1/CCR3, rs4744112: ROR2/SPTLC1, rs10840031: TRIM66, rs800139: C11orf21, rs1518110: IL10, and rs56350303: AC091814.2. These indicate the potential functional genes behind the identified genetic loci.

## Discussion

The present study applied a high-throughput functional genomics approach to identify the associations between genetic factors and inflammatory phenotype in patients with T1D. The results confirm a correlation between baseline immune-cell populations and ex vivo cytokine production in response to bacterial, fungal, non-microbial, and TLR ligand stimulations. We provide evidence for a direct link between T1D GWAS loci and immune functionality, particularly through circulating T cell subpopulations. We show that T cell alteration is largely driven by T1D genetics, while B cells do not show a significant association with T1D GWAS loci. The association between the proportion of CCR5+ Tregs and T1D susceptibility through CCR genes suggests that T1D-associated genetic variants contribute to alteration of immune function through a cumulative effect. Finally, out of 28 genome-wide significant

genetic loci regulating immune-cell proportions and cytokine production, we identified 12 immune phenotype QTLs specific to 300DM. We also found 11 druggable genes as candidates for therapeutic intervention. Altogether, this study provides several novel insights into the genetic variability of immune traits in T1D.

The correlation we found between baseline immune cells and cytokine production in response to bacterial, fungal, non-microbial, and TLR ligand stimulations suggests that steady-state immune-cell abundance and alteration of immune-cell proportion also affect immune response. This is in line with earlier findings on the regulation between immune cells and cytokine production (*Kim et al., 2007*). This finding has two important consequences: (1) immune response variability could be partly explained by variations in steady state and (2) treatment of steady-state immune cells before stimulation could also influence immune response after stimulation.

Studies comparing T1D patients and healthy controls show that immune-related genes are physically located in T1D loci or are differentially expressed. More studies highlight the importance of T cell immunity in T1D pathology (*Farh et al., 2015*). In our study, we provide evidence for a direct link between T1D GWAS loci and immune functionality, particularly for circulating T cell subpopulations. We show that T cell alteration is largely genetically driven, while, interestingly, B cells do not show a significant association with T1D GWAS loci. It is thus tempting to speculate that the occurrence of auto-antibodies in T1D patients has limited pathophysiological relevance and is merely a marker of an autoimmune process mainly driven by T cells, as has been suggested before (*Martin et al., 2001*).

Importantly, we reveal that the effects of different GWAS loci on immune phenotypes vary. Despite the strong effects of the CCR locus (mean absolute effect sizes = 0.017 [cell proportion] and 0.15 [cytokine production]), we observed much weaker individual effects for other T1D GWAS loci on immune-cell proportion parameters (mean absolute effect size = 0.0060) and/or cytokine production (mean absolute effect size = 0.10) (*Figure 2C* and *Figure 2—figure supplement 2*). This finding indicates that T1D-associated genetic variants might alter immune function through a cumulative effect. Considering the complexity of the immune system, this may be an alternative explanation for our finding that T1D-associated genetic variants that affect immune functions differ from the genetic variants that affect immune-cell proportion and cytokine production capacity.

Despite the limited sample size, we identified 28 genome-wide significant genetic loci regulating immune-cell proportions and cytokine production in T1D patients and health. Among them, we found 12 immune phenotype QTLs in 300DM but not in healthy volunteers, suggesting a distinct regulatory mechanism of immune parameters and functions in disease compared to health. More importantly, our results highlight 11 druggable genes as candidates for therapeutic intervention.

The data presented in our study were generated from PBMC. While these likely reflect overall immune function, some immune-cell types may not be captured and the findings refer to changes in circulating factors that may not necessarily reflect changes occurring in relevant immune organs such as pancreatic islets, gut, or lymph nodes. Nonetheless, islet-infiltrating immune cells do originate from circulating blood cells, while circulating chemokines/cytokines are important in activating and recruiting immune cells. Hence, the circulating level of immune cells and cytokine production capacity are probably relevant for local tissue immunity.

We acknowledge that our study has limitations. First, 300DM and 500FG were recruited and measured 2 years apart, albeit in an identical setting and following the same protocol in the same lab. There may thus be some differences in the absolute immune-cell counts or cytokine levels due to a batch effect. Therefore, this study was not designed as a case-control study, but the healthy controls were used to compare genetic associations identified in the T1D cohort. Second, young people are overrepresented in 500FG. Although we regressed age out in both cohorts, there may still be a bias in the genetic mapping. Third, our investigation on circulating cytokines in this study was focused on the variation of inflammatory responses, which are mediated by innate immune cells and are antigen-independent. Future study could focus on the stimulation of PBMC with beta-cell autoantigens such as insulin or GAD65 peptides, to assess specific T and B cell functional responses. Finally, our T1D-specific analyses should be viewed as exploratory because they have not been validated in a separate cohort. Our study has also strengths. We applied cutting-edge technologies to assess immune-cell function and genetic variation, and this is the first study to comprehensively combine 'omics' technologies with abundant phenotyping in a rather large group of participants (N=215) to explain intra-individual variation in immune responses. Moreover, we stimulated mononuclear cells with ligands of

pattern recognition receptors, such as TLR2 and TLR4, known to induce effective inflammation. These stimulations yield significant releases of cytokines allowing detailed quantification of various inflammatory pathways in each subject.

In conclusion, by applying a novel high-throughput functional genomics approach, we have shown that genetic factors regulate immune responses in T1D patients. We show genetic susceptibility to immune phenotypes in patients with T1D and highlight the importance of T cell immunity in the genetic regulation of T1D. We also identify specific cell populations (CCR5+ Tregs) that are likely involved in pathophysiology of T1D. Together, these findings may provide an avenue toward identification of novel preventive and therapeutic treatments.

## Data and code availability

All the raw data on immune phenotypes and summary statistics generated directly from genetic data needed to precisely reproduce published results are deposited in Dryad (https://doi.org/10.5061/dryad.4f4qrfjd0). Custom scripts for generating summary statistics and all results are deposited in GitHub (https://github.com/Chuxj/Gf_of_ip_in_T1D, copy archived at swh:1:rev:1e39df29db-f38a94b9e2325827ac94043d190be7; *Chux, 2021*). Individual genetic data and other privacy-sensitive individual information are not publicly available because they contain information that could compromise research participant privacy. For data access, please contact Prof. Cees Tack ( Cees.Tack@radboudumc.nl). This original data is available for qualified researchers, that is, senior investigators employed or legitimately affiliated with an academic, non-profit, or government institution who have a track record in the field. We would ask the researcher to sign a data access agreement that needs to be signed by applicants and legal representatives of their universities. In addition, we would require a research proposal, to ensure that 'Applications for access to Data must be Specific, Measurable, Attainable, Resourced and Timely.' The applicant must implement the proposed research within the designed time frame and the data must be deleted after finishing the proposal.

## Acknowledgements

We thank all of the volunteers in the 300DM and 500FG for their participation. We thank Marc J Bonder for discussions and Kate Mc Intyre for editorial work.

## Additional information

### Funding

| Funder | Grant reference number | Author |
|---|---|---|
| ERC Starting grant | 948207 | Yang Li |
| Radboud Universitair Medisch Centrum | Hypatia Grant 2018 | Yang Li |
| ERC advanced grant | 833247 | Mihai G Netea |
| the Netherlands Association of Scientific Reasearch | Spinoza grant | Mihai G Netea |
| the Netherlands Organisation for Scientific Research | Perspectief Biomarker Development Center Research Programme | Cees J Tack |
| European Foundation for the Study of Diabetes | AZ Macrovascular Programme 2015 | Anna WM Janssen |
| China Scholarship Council | 201706040081 | Anna WM Janssen |

The funders had no role in study design, data collection and interpretation, or the decision to submit the work for publication.

## Author contributions
Xiaojing Chu, Conceptualization, Data curation, Formal analysis, Investigation, Methodology, Visualization, Writing - original draft; Anna WM Janssen, Data curation, Resources, Writing – review and editing; Hans Koenen, Conceptualization, Resources, Writing – review and editing; Linzhung Chang, Formal analysis, Validation; Xuehui He, Formal analysis, Resources; Irma Joosten, Rinke Stienstra, Resources, Writing – review and editing; Yunus Kuijpers, Validation, Writing – review and editing; Cisca Wijmenga, Mihai G Netea, Resources, Supervision, Writing – review and editing; Cheng-Jian Xu, Supervision, Writing – review and editing; Cees J Tack, Funding acquisition, Project administration, Supervision, Writing – review and editing; Yang Li, Conceptualization, Funding acquisition, Project administration, Supervision, Writing – review and editing

## Author ORCIDs
Xiaojing Chu (iD) http://orcid.org/0000-0002-9882-2912
Yunus Kuijpers (iD) http://orcid.org/0000-0002-5075-3970
Cisca Wijmenga (iD) http://orcid.org/0000-0002-5635-1614
Cheng-Jian Xu (iD) http://orcid.org/0000-0003-1586-4672
Mihai G Netea (iD) http://orcid.org/0000-0003-2421-6052
Yang Li (iD) http://orcid.org/0000-0003-4022-7341

## Ethics
Human subjects: The 500FG-DM study was approved by the ethical committee of Radboud University Nijmegen (NL-number: 54214.091.15). Experiments were conducted according to the principles expressed in the Declaration of Helsinki. Written informed consent was obtained from all participants.

## Decision letter and Author response
Decision letter https://doi.org/10.7554/eLife.73709.sa1
Author response https://doi.org/10.7554/eLife.73709.sa2

---

# Additional files

## Supplementary files
• Supplementary file 1. Genetic mapping and summary statistics of immune phenotypes in type 1 diabetes (T1D). (A) Definition of parental and grandparental cell types. (B) Numbers and basic characteristics of participants included in the study. (C) Summary statistics of reported T1D genome-wide association studies (GWAS) single-nucleotide polymorphisms (SNPs) in 300DM and 500FG. (D) Summary statistics of GWAS loci in cell proportion quantitative trait loci (QTLs) profile. (E) Summary statistics of GWAS loci in cytokine QTLs profile. (F) Genomic loci identified in T1D and meta-analysis.

• Transparent reporting form

## Data availability
All the raw data on immune phenotypes and summary statistics generated directly from genetic data needed to precisely reproduce published results are deposited in Dryad (https://doi.org/10.5061/dryad.4f4qrfjd0). Custom scripts for generating summary statistics and all results are deposited in GitHub (https://github.com/Chuxj/Gf_of_ip_in_T1D, copy archived at swh:1:rev:1e39df29dbf38a94b9e2325827ac94043d190be7). Individual genetic data and other privacy-sensitive individual information are not publicly available because they contain information that could compromise research participant privacy. For data access, please contact Prof. Cees Tack (Cees.Tack@radboudumc.nl). This original data is available for qualified researchers, i.e. senior investigators employed or legitimately affiliated with an academic, non-profit or government institution who have a track record in the field. We would ask the researcher to sign a data access agreement that needs to be signed by applicants and legal representatives of their universities. In addition, we would require a research proposal, to ensure that 'Applications for access to Data must be Specific, Measurable, Attainable, Resourced and Timely.' The applicant must implement the proposed research within the designed time frame and the data must be deleted after finishing the proposal.

The following datasets were generated:

| Author(s) | Year | Dataset title | Dataset URL | Database and Identifier |
|---|---|---|---|---|
| Chu X, Li Y, Tack C, Janssen A, Koenen H | 2021 | Datasets for Genetic and environmental effects on the immune phenotypes in type 1 diabetes | https://doi.org/10.5061/dryad.4f4qrfjd0 | Dryad Digital Repository, 10.5061/dryad.4f4qrfjd0 |
| Chu X | 2021 | Custom scripts for Genetic and environmental effects on the immune phenotypes in type 1 diabetes | https://github.com/Chuxj/Gf_of_ip_in_T1D | GitHub Repository, Chuxj/Gf_of_ip_in_T1D |

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
