## [Editor Report]

This study examines genetic and non-genetic factors influencing immune responses in type 1 diabetes Key findings are: 1) age and season affect immune cell traits and cytokine production upon stimulation; 2) certain genetic variants that determine susceptibility to T1D significantly affect T cell composition, notably the CCR region that is associated with CCR5+ regulatory T cells; and 3) 15 genetic loci that influence immune responses in T1D, most of which have not been seen previously in healthy populations. The results suggest mechanisms of T1D-specific genetic regulation.

---

## [Decision Letter]

**Decision letter after peer review:**

Thank you for submitting your article "Genetic and environmental effects on the immune phenotypes in type 1 diabetes" for consideration by *eLife*. Your article has been reviewed by 3 peer reviewers, and the evaluation has been overseen by a Reviewing Editor and Mone Zaidi as the Senior Editor. The following individual involved in review of your submission has agreed to reveal their identity: Peter S Linsley (Reviewer #1).

Essential revisions:

While the reviewers acknowledged the potential value of this work, they criticized that the paper lacked focus and needed significant editing and rewriting.

1. Please address the many requests for clarification and/or more nuanced discussion as outlined by the reviewers.

2. Define the population structure and perform an analysis for population stratification.

3. Define how the 1000 random SNPs were selected.

4. Describe the methodology of meta-analysis used in more detail.

5. Provide figure legends that summarize the results rather than the tasks executed.

6. Focus on the genetic association with immune phenotypes in your analysis.

[Editors’ note: further revisions were suggested prior to acceptance, as described below.]

Thank you for resubmitting your work entitled "A genome-wide functional genomics approach uncovers genetic determinants of immune phenotypes in type 1 diabetes" for further consideration by *eLife*. Your revised article has been evaluated by Mone Zaidi (Senior Editor) and a Reviewing Editor.

Could the authors discuss the limitation that the stimulation of PBMC's was done using generic stimulants and not β-cell autoantigens such as insulin or GAD65 peptides. This would address Reviewers 2 criticism.

*Reviewer #1 (Recommendations for the authors):*

This revised manuscript has incorporated extensive modifications to the original. The modifications have improved the manuscript and I have no further major objections to publication.

*Reviewer #2 (Recommendations for the authors):*

Most comments were addressed.

However, an important critique was not addressed or commented on by the authors, and that is the fact that stimulation of PBMC's was done using generic stimulants and not β-cell autoantigens such as insulin or GAD65 peptides. This weakness of the work needs to be addressed.

*Reviewer #3 (Recommendations for the authors):*

The authors have substantially improved this manuscript by re-focusing the data that is presented and discussed on their most interesting findings. Generally, the manuscript is now much easier to read and follow, and it makes important points. The discussion in particular is well composed and acknowledges both strengths and weaknesses of the study.

---

## [Author Response]

Essential revisions:While the reviewers acknowledged the potential value of this work, they criticized that the paper lacked focus and needed significant editing and rewriting.1. Please address the many requests for clarification and/or more nuanced discussion as outlined by the reviewers.

Thank you for the suggestion. We have at several places provided details and clarification according to the suggestions, as pages 17-20 of the revised manuscript show.

2. Define the population structure and perform an analysis for population stratification.

We have provided required details about the population used in this study and have performed the suggested analysis for population stratification. The results have been added to the section “Genotyping, quality control and imputation”, lines 139-146 of the revised manuscript and in the Figure 1—figure supplement 1. Details are also listed below:

“The genotype calling was performed using Opticall 0.7.0 (Shah et al., 2012) with default settings (call rate > 0.99). Two samples were removed due to contamination or mislabeling; one sample from a related individual (identity be descent > 0.185) was removed and six genetic outliers were identified by either heterozygosity rate check (individuals with heterozygosity rate ± 3 standard deviations from the mean were excluded) or multi-dimensional scaling (MDS) plots of samples merging with 1000 Genomes data, leaving 215 samples, which show good consistence with European samples from 1000 Genomes data (Figure 1—figure supplement 1) according to standard protocol (Anderson et al., 2010).”

3. Define how the 1000 random SNPs were selected.

We agree with the reviewers that it is relevant to include this information in the manuscript. We have updated the manuscript on lines 200-208 (lines 577-585 in the tracking changes version), as shown below.

“In order to detect the impact of T1D GWAS loci on immune cell populations, we grouped all traits into four categories (B cells, T cells, monocytes and NK cells), and counted the number of suggestive associations (P value < 0.05) between 63 top SNPs from T1D GWAS loci and immune cell traits. 1000 permutated sets of 63 SNPs were randomly selected from independent SNPs (r2 > 0.2) pruned from all genotyped SNPs. We then compared the associations of the 63 top GWAS SNPs with associations between 1000 permuted sets and the same category of immune traits. P value was calculated by the percent of association numbers from permuted sets greater than the association number of the 63 T1D GWAS SNPs.”

4. Describe the methodology of meta-analysis used in more detail.

We apologize for the missing detailed information. We have now added the details of meta-analysis in the *Method* sections on lines 182-185 (lines 556-558 in tracking changes version), as the following text shows:

“……, and software METAL(Willer et al., 2010) was applied to summary statistics from both cohorts in the meta-analysis, in which the model based on effect size and standard error with default settings were used. P values < 5×10^-8^ were considered to be genome-wide significant.”

5. Provide figure legends that summarize the results rather than the tasks executed.

We thank the reviewers for the suggestion. We have now modified the figure legends by adding text for describing the results.

“Figure 1. Overview of data and experimental design. A, Schematic study design. B,C, Cell (B) and cytokine (C) type relationship visualized by non-metric multidimensional scale (NMDS) analysis in 300DM, where B cell traits (red) are clustered separately from T cell and monocyte traits (B) and cytokines with same cellular origins are clustered together (C). In panel B, Small dots indicate the proportion of subpopulations and large dots indicate counts of their parental cell types. Circles represent the calculated centroid of the grouped cell and cytokine types at confidence level 0.95. D, Heatmap of correlation coefficients between immune cell counts (y-axis) and cytokine production in response to stimulations (x-axis) in T1D patients. Significant correlations (FDR <0.05) are labeled by dots, with the color indicating correlation coefficients. Positive correlations (red) are observed between monocyte traits and monocyte-derived cytokines (top left), and between adaptive immune cell counts and T cell-derived cytokines (bottom right). Negative correlations (blue) are observed between adaptive immune cell counts and monocyte-derived cytokines.”

“Figure 2. Impact of T1D GWAS SNPs on immune phenotypes. A, QQ plots of QTL profiles of 62 T1D GWAS loci grouped by cell populations. The distribution of P values of associations with T cells traits (blue) show a significant deviation from an expected uniform distribution (dashed line). B, Histogram showing number of associations observed (red line) and those in permutations (blue bars). C, Heatmap of QTL profiles of cell proportion carrying certain chemokine receptors across 62 T1D GWAS loci, colored by -log10(p values) and effect direction of T1D risk allele. Arrowhead indicates a T1D risk allele rs11574435-T.”

“Figure 3. Impact of T1D GWAS SNPs on immune phenotypes. A, Expression of CCR5, CCL5 and CCL4 in PBMCs (top) and Pancreases (bottom) altered in T1D patients (blue, green and purple) compared to healthy controls (red). B, A locus zoom plot showing SNPs around rs35092096 located in CCR region are associated with CCR6+CCR7-CCR5+ T reg proportion. C, A boxplot showing CCR6+CCR7-CCR5+ T reg proportion differs in different rs35092096 genotypes (TT: red, TG: green and GG: blue). D, Two locus zoom plots indicating colocalization between CCR6+CCR7-CCR5+ T reg proportion QTL profiles (top, red) and T1D GWAS profile (bottom, blue) within CCR regions.”

“Figure 4. Genetic regulators on immune phenotypes. A, Three Manhattan plots showing genetic regulators of immune cell proportion (top), proportion of immune cells expressing CCR (middle) and cytokines production in response to stimulations (bottom). P values of SNPs identified in meta-analysis were colored in red. B, A locus zoom plot showing a T1D specific regulatory locus around rs4744112 that effects CCR6+CXCR3+CCR4- T helper1 proportion. C, A boxplot showing CCR6+CXCR3+CCR4- T helper1 proportion varies in different rs4744112 genotypes (TT: red, TG: green and GG: blue).”

6. Focus on the genetic association with immune phenotypes in your analysis.

Thanks for the suggestion. We have decided to take the non-genetic associations, including seasonal effects and age, completely out of the paper and have rewritten the paper accordingly. Hence also one figure was removed, the others were renumbered.

[Editors’ note: further revisions were suggested prior to acceptance, as described below.]

Reviewer #2 (Recommendations for the authors):Most comments were addressed.However, an important critique was not addressed or commented on by the authors, and that is the fact that stimulation of PBMC's was done using generic stimulants and not β-cell autoantigens such as insulin or GAD65 peptides. This weakness of the work needs to be addressed.

The reviewer is correct that we haven’t addressed this comment in detail in our earlier revision, our excuses. The aspect of the type of stimulus used is indeed very important and depends on the scientific question asked. If our focus would have been the assessment of T- and B-cell specific responses, stimulation with β-cell autoantigens such as insulin or GAD65 peptides would have been indeed important. However, the focus of our investigation was to assess variation in inflammatory responses, which are mediated by innate immune cells (such as monocytes and macrophages) and are antigen-independent. The direct stimulation of PBMCs (mainly lymphocytes and monocytes) with for example GAD65 will yield a very weak response if at all measurable and will probably not be informative about functionality of myeloid cells. On the other hand, we stimulated mononuclear cells with ligands of pattern recognition receptors, such as TLR2 and TLR4, known to induce effective inflammation. These stimulations yield significant releases of cytokines allowing detailed quantification of various inflammatory pathways in each subject. Experiments to assess specific T- and B-cell function have not been performed in our cohort, and we now acknowledge and discuss this topic in more detail in the revised version.

In line 439-442:

Thirdly, our investigation on circulating cytokines in this study was focused on the variation of inflammatory responses, which are mediated by innate immune cells and are antigen-independent. Future study could focus on the stimulation of PBMC with β-cell autoantigens such as insulin or GAD65 peptides, to assess specific T- and B-cell functional responses.

And in line 448-451:

Moreover, we stimulated mononuclear cells with ligands of pattern recognition receptors, such as TLR2 and TLR4, known to induce effective inflammation. These stimulations yield significant releases of cytokines allowing detailed quantification of various inflammatory pathways in each subject.